# Restoring brain connectivity by phrenic nerve stimulation in sedated and mechanically ventilated patients
Thiago Bassi [1,2] ✉, Elizabeth Rohrs E[3], Melodie Parfait[4], Brett C. Hannigan [5], Steven Reynolds [3], Julien Mayaux[4], Maxens Decavèle[4], Jose Herrero [6,7], Alexandre Demoule[4], Thomas Similowski[8] & Martin Dres [4] ✉

## Abstract

**Background** In critically ill patients, deep sedation and mechanical ventilation suppress the brain-diaphragm-lung axis and are associated with cognitive issues in survivors.
**Methods** This exploratory crossover design study investigates whether phrenic nerve stimulation can enhance brain activity and connectivity in six deeply sedated, mechanically ventilated patients with acute respiratory distress syndrome.
**Results** Our findings indicate that adding phrenic stimulation on top of invasive mechanical ventilation in deeply sedated, critically ill, moderate acute respiratory distress syndrome patients increases cortical activity, connectivity, and synchronization in the frontal-temporal-parietal cortices.
**Conclusions** Adding phrenic stimulation on top of invasive mechanical ventilation in deeply sedated, critically ill, moderate acute respiratory distress syndrome patients increases cortical activity, connectivity, and synchronization. The observed changes resemble those during diaphragmatic breathing in awake humans. These results suggest that phrenic nerve stimulation has the potential to restore the brain-diaphragm-lung crosstalk when it has been shut down or impaired by mechanical ventilation and sedation. Further research should evaluate the clinical significance of these results.

## Plain language summary

Critically ill patients receive sedation and mechanical ventilation as life support measures. Sedation and mechanical ventilation impact the ability of the brain, lungs and diaphragm to communicate as normal. We studied whether stimulating a nerve involved in breathing, called the phrenic nerve, would lead to brain activity in deeply sedated and mechanically ventilated patients. Stimulating the phrenic nerve increased the amount of communication within the brain and the communication seen was similar to that seen in studies in healthy participants. These data suggest that stimulating the phrenic nerve could restore the brain-diaphragm-lung crosstalk when it has been shut down or impaired by mechanical ventilation and sedation which could prevent cognitive impairment in critically ill patients.

In the intensive care unit (ICU), patients receiving mechanical ventilation (MV) are often deeply sedated, especially in the case of acute respiratory distress syndrome (ARDS)[1]. While sedation and MV are life-saving interventions they come with serious risks, including ICU delirium and post-ICU cognitive impairment[2,3]. In animal models, MV is associated with hippocampus apoptosis and brain inflammation[4–8]. These drawbacks of sedation and MV underlie the recent clinical concept of ventilator-associated brain injury (VABI), which may be a direct consequence of abolishing the brain-lung crosstalk[4]. Recent

studies in animals have shown that restoring brain-lung crosstalk via diaphragm contractions induced by phrenic nerve stimulation can prevent VABI[6,8,9].

The phrenic nerve provides motor innervation to the diaphragm, the primary muscle responsible for resting breathing[10]. It also conveys the diaphragm's sensory innervation, and previous studies have shown that phrenic nerve stimulation recruits phrenic afferent fibers[10] that project to the parietal diaphragmatic (sensory) cortex and the limbic system[10–15].

[1]Lungpacer Medical Inc., Vancouver, BC, Canada. [2]Interdepartmental Division of Critical Care Medicine, University of Toronto, Toronto, ON, Canada. [3]Advancing Innovation in Medicine Institute, New Westminster, BC, Canada. [4]Sorbonne Université, INSERM, UMRS1158 Neurophysiologie respiratoire expérimentale et clinique, AP-HP. Sorbonne Université, Hôpital Pitié-Salpêtrière– Service de Médecine Intensive et Réanimation, Paris, France. [5]ETH Zurich, Department of Health Sciences and Technology, Zurich, Switzerland. [6]The Feinstein Institutes for Medical Research, Northwell Health, New York, NY, 11030, USA. [7]Hofstra Northwell School of Medicine, New York, NY, 11549, USA. [8]Sorbonne Université, INSERM, UMRS1158 Neurophysiologie respiratoire expérimentale et clinique, AP-HP. Sorbonne Université, Hôpital Pitié-Salpêtrière– (Département "R3S"), F-75013 Paris, France. ✉e-mail: thiago.gasperinibassi@uhn.ca; martin.dres@aphp.fr

In awake, healthy humans, diaphragmatic breathing – a breathing technique favoring the diaphragm over other inspiratory muscles to produce inspiration – modifies the electroencephalogram (EEG) spectral activity, with increased gamma and alpha power in frontal-temporal-parietal regions[16–19]. Additionally, diaphragmatic breathing changes brain perfusion[20,21], improves cerebrospinal fluid dynamics[22], and enhances cognitive performance[20,21].

From these observations, we reasoned that phrenic stimulation, via a central venous catheter[23,24], could restore the brain-diaphragm-lung cross-talk in sedated MV ARDS patients, with effects resembling those of diaphragmatic breathing in awake subjects. We therefore hypothesized that, in ARDS patients, phrenic stimulation would induce EEG spectral changes involving the alpha and gamma bands in the frontoparietal cortical regions and would activate the limbic system. We further hypothesized that phrenic stimulation would modify cortical synchronization and connectivity between brain areas as well as the autonomic control of brain vessels, as assessed via functional near-infrared spectroscopy (fNIRS). We tested these hypotheses by conducting exploratory brain measurements in ARDS patients in a study that investigated the effects of phrenic nerve stimulation on lung mechanics[25].

We found that phrenic nerve stimulation increases cortical activity, connectivity, and synchronization in critically ill mechanically ventilated patients. The observed changes resemble those during diaphragmatic breathing in awake humans.

## Methods

This report is an ancillary investigation from a cross-sectional single-center study (ClinTrials.gov: NCT04844892) approved by the ethics committee (Comité de Protection de Personnes (CPP), Ile-De-France X, Paris, France) on June, 15th, 2021. The original study was designed to investigate the effects on lung mechanics of phrenic nerve stimulation in critically ill mechanically ventilated patients. The results (lung mechanics, hemodynamics and gas exchange) have been published elsewhere[25] and the present study reports unpublished data of a subgroup of patients whose brain data were collected after an amendment (21.04819.492021-MS01) to the primary study was approved on February, 15th, 2022 by the same ethics committee. Eligible patients were admitted to a 22-bed medical ICU at the APHP Sorbonne Université Pitié Salpêtrière Hospital, Paris, France. A signed informed consent was obtained from the substitute decision-makers (legally people who gave informed consent on behalf of patients) for each patient prior to study inclusion. The first author of the study is affiliated with the study sponsor, Lungpacer Medical Inc., which however had no role as an entity in the data analyses or manuscript writing. *Inclusion criteria* were patients with moderate ARDS according to the Berlin definition[26], receiving invasive MV for at least 48 h but no longer than 5 days, and receiving continuous sedation, with Richmond Agitation Sedation Scale (RASS) ≤−3. *Exclusion criteria were:* septic shock with hemodynamic instability (requiring norepinephrine or epinephrine >0.5 μg/kg/min); impossible access to the left subclavian vein; use of neuromuscular blocking agents within the last 12 h; bacteremia documented within the last 48 h or uncontrolled source of infection; ongoing extracorporeal lung circulation; enrolled in other studies of an investigational drug or device that could have affected the outcomes of the current study; pre-existing neurological; neuromuscular or muscular disorder that could have affected the respiratory muscles; body mass index >45 kg/m²; known or suspected phrenic nerve paralysis; any electrical device (implanted or external) that could have been prone to interaction with or interference from the phrenic stimulation catheter, including neurological pacing/stimulator devices and cardiac pacemakers and defibrillators; no affiliation to the French health insurance system, curatorship, imprisoned, known or suspected to be pregnant, or lactating. Thus, none of the enrolled patients suffered from phrenic nerve paralysis.

### Study design

All patients had a central-line catheter embedded with electrodes (Lungpacer Medical Inc., Canada) inserted into either the left subclavian vein or left internal jugular vein. This was used to stimulate the phrenic nerves bilaterally, in synchrony with MV. The study protocol consisted of a series of four consecutive 60 min sessions, with sessions 1 and 3 unpaced and sessions 2 and 4 paced (Fig. 5). The first and third sessions were periods of standard-of-care protective lung ventilation alone whereas in the second and fourth sessions each respiratory cycle resulted from stimulation of the phrenic nerve, either "ventilator triggered" (ie, the ventilator initiated the breathing cycle) or "pacing triggered" (ie, the pacer initiated the breathing cycle) in combination with standard-of-care lung protective ventilation (Fig. 5). Ventilator settings and continuous medications were not modified during the study.

**Delivery of phrenic nerve stimulation in conjunction with mechanical ventilation.** Lung protective ventilation was applied as a standard of care at inclusion and during all sessions. Lung protective ventilation was delivered in volume control mode with tidal volume set between 6 and 8 mL/kg of predicted body weight, positive end-expiratory pressure (PEEP) set to maintain plateau pressure <30 cmH₂O, and respiratory rate set to achieve pH between 7.35 and 7.45. A maximal respiratory rate of 35 breaths/minute was allowed. After checking the correct placement of the catheter using chest X-ray, mapping was performed to select appropriate pairs of electrodes to enable right and left diaphragm recruitment[6,25,27,28]. Diaphragm recruitment was confirmed by checking the occurrence of sudden saccadic deflections of the airway pressure waveform on the ventilator screen. Phrenic nerve stimulation was delivered using a standalone stimulator (Lungpacer Medical Inc., Vancouver, Canada) that was connected to an adapted airflow sensor between the endotracheal tube and the ventilator "Y" connector, to detect inspiration and allow synchronization with the ventilator.

Phrenic nerve stimulation was set and titrated to achieve up to a 15% reduction of the ventilator pressure time product, based on previous studies[25,27,28]. The pressure time product was continuously monitored on a laptop (Fluxmed system, GrE, L3 Medical). All lung physiology measurements from this pilot study have been previously reported[25]. Phrenic nerve stimulation was delivered as trains of pulses with a pulse frequency of 40 Hz and a pulse width between 200 and 300 microseconds. Stimulation trains were set to be 100 microseconds shorter than the inspiratory time set on the ventilator. The maximum total current delivered could not exceed 27 mA via either bipolar or multipolar stimulation.

**Data acquisition.** Cardiac index was recorded at 5 Hz by a transpulmonary thermodilution device (PiCCO₂, Pulsion Medical Systems, Feldkirchen, Germany). An EEG (SAGA 32+ TMSi SAGA, The Netherlands) and a fNIRS (BRITE Artinis Medical Systems, The Netherlands) device were used concomitantly to collect brain signals and changes in oxyhemoglobin and deoxyhemoglobin, respectively. The EEG and fNIRS data were collected and stored by using OxySoft software (Artinis Medical Systems, The Netherlands). EEG and fNIRS signals were processed with MATLAB (MATLAB Version: 9.13.0-R2022b) by one of the authors (BH) blinded to the session allocations.

**EEG.** A 32-channel EEG with a sampling rate of 2000 Hz and a notch filter centered at 50 Hz was used to collect brain activity from 28 channels. The activity from four occipital channels was not collected due to ICU bed-space limitations and to avoid unnecessary mobilization of the patient's head. The electrodes were placed in a head cap according to the 10–20 electrode placement system[29]. The combination of EEG electrodes and their respective anatomical projections are presented in Fig. 5A.

After applying a 40–60 Hz notch, EEG data were analyzed by comparing paced versus unpaced sessions. Brain connectivity analysis was done via a multicorrelation matrix.

**EEG power spectral analysis.** The EEG power spectral analysis was conducted by comparing the effect observed during paced versus unpaced sessions. Data were pooled and averaged using the first and last

10 min of each session for each patient, to compare signals obtained from the frontal, temporal, and parietal lobes during paced and unpaced sessions (Fig. 5, bottom). The EEG signals were then processed by Fourier transformation (Matlab) to calculate the power spectrum density (PSD) expressed as $\mu V^2$/Hz for all standard EEG frequency bands: delta (0.5–4 Hz), theta (4–8 Hz), alpha (8–12 Hz), beta (13–30 Hz), and gamma (30–100 Hz).

**EEG connectivity analysis.** Brain connectivity analysis was conducted for each EEG frequency band by constructing a correlation matrix using the Matlab EEGLAB toolbox[30]. The gamma frequency band was subdivided into low (30–50 Hz), mid (50–70 Hz), and high (70–100 Hz) bands because of its larger frequency range[18,31]. After constructing a correlation matrix with data from the unpaced and paced sessions, edges were defined as absolute values of the Pearson correlation coefficient applied to all pairwise electrodes without lag[32]. A correlation matrix was then generated with Pearson correlations. The correlation coefficient threshold of 0.4 was used to investigate any correlation stronger than moderate, following published literature[32]. The edges with a below-threshold correlation value were removed[32]. The global efficacy method was used to test the connections between the brain areas where data were collected (frontal, temporal, and parietal) by measuring the overall capacity to transfer and integrate the information between brain areas[19]. Part of the global efficacy measures the minimum number of edges needed to connect two or more areas of the brain. A higher number of edges indicates anatomically and neurophysiologically related firing patterns, suggesting high connectivity[19]. Brain connectivity was graphed in a circular diagram displaying relationships between pairs of regions (Fig. 4)[32]. All p-values of the differences in brain connectivity between the paced and unpaced sessions that had a correlation coefficient greater than 0.4 were reported in a heatmap.

**fNIRS system.** A 28-channel functional near-infrared system was used to identify changes in oxyhemoglobin and deoxyhemoglobin in the frontal cortexes. Data from other brain regions were not collected because the number of optodes was not sufficient to cover the whole cranium. A multimodal placement system was used to integrate simultaneously EEG plus fNIRS placement in the head of the patients (Fig. 5B)[33]. The variables of interest were oxyhemoglobin and deoxy-hemoglobin tissue concentrations as well as variations in the amplitude of the oscillations of the near-infrared signal that constituted "frequency signatures" from various brain structures[34–36]. These" frequency signatures" consist of low-frequency oscillations (LFO, ranging from 0.09 to 1.25 Hz) and very low-frequency oscillations (VLFO, ranging from 0.08 to 0.01 Hz)[34–36]. LFO and VLFO amplitudes are expressed in nanovolts (nV) due to the electromagnetic proprieties of light[34–36]. According to published literature, oscillation amplitude changes observed between 1.0 and 1.25 Hz are related to cardiac rhythm, amplitude changes between 0.18 and 0.25 Hz range are related to breathing activity, amplitude changes between 0.08 and 0.13 Hz are related to the myogenic frequency (sympathetic tone), amplitude changes between 0.05 and 0.07 Hz are related to the neurogenic frequency (large-size artery tonus), and amplitude changes between 0.01 and 0.03 Hz are related to the metabolic frequency (small vessel or capillary tonus)[34,35].

**Statistics and reproducibility.** Linear data analysis was done using a non-parametric Friedman paired test with the Bonferroni correction for multiple comparisons to compare brain activity for each EEG/fNIRS

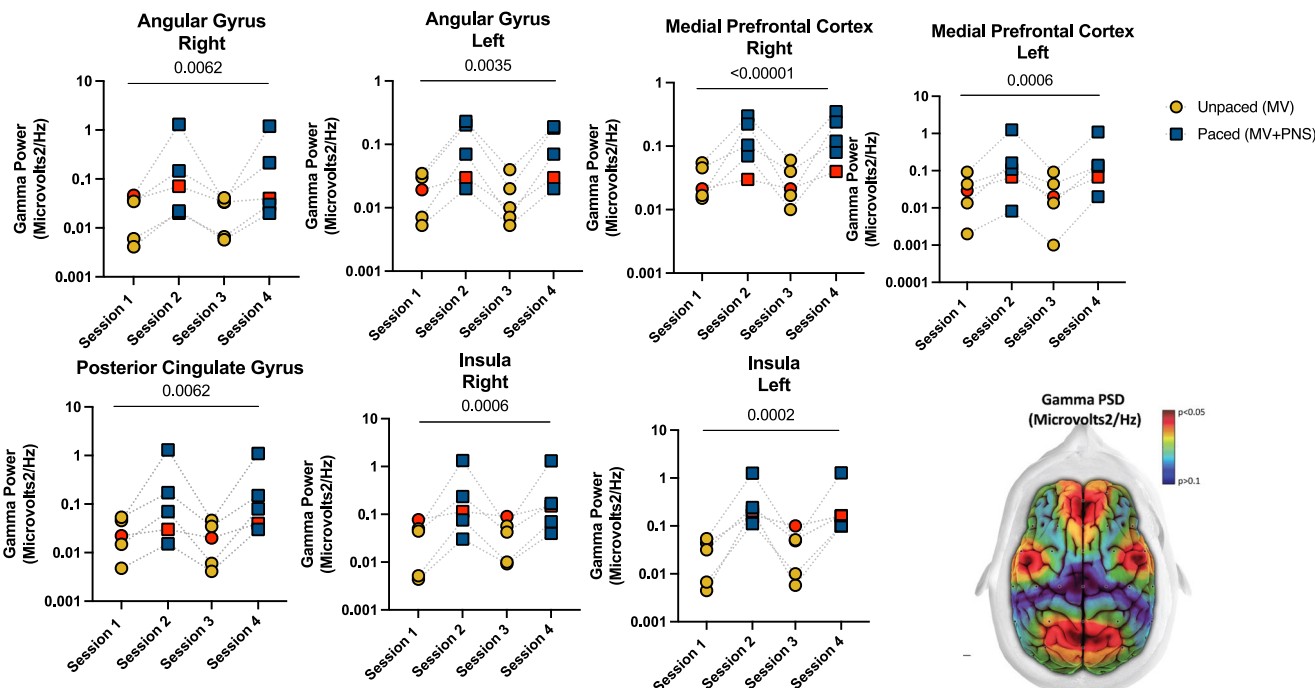

**Fig. 1 | Phrenic nerve increases EEG gamma frequency in deeply sedated mechanically ventilated patients.** EEG gamma power differences between paced and unpaced sessions: Dot plots show the mean EEG gamma frequency (30–100 Hz) power spectrum in brain regions where statistically significant differences between sessions were observed. The yellow and blue dots represent unpaced (MV, mechanical ventilation) and paced (MV + PNS, mechanical ventilation plus phrenic nerve stimulation) sessions respectively. The red dots show the data from Patient #1 who had an out-of-the-hospital cardiac arrest and was diagnosed with acute hypoxic-ischemic encephalopathy (a possible negative control due to an impaired brain-lung crosstalk). On the bottom right, a brain topoplot displays the anatomical brain regions where a statistically significant change was observed between paced vs. unpaced sessions. The red color in the topoplot indicates statistical significance defined as $p < 0.05$. The gradient of colors from yellow to dark blue indicates p-values greater than 0.05. (Source data file in Supplementary Data 1).

signal between sessions. A subgroup analysis using a mixed-effect analysis of the variance of the magnitude effect of the phrenic nerve stimulation on the brain was conducted to investigate whether patients' characteristics could modulate the magnitude of the observed effect during the phrenic nerve stimulation sessions. Patients were divided into three subgroups, hypoxic-ischemic (#1) encephalopathy, non-drug addicted (#2, #4, #6), and drug addicted (#3, #5).

Data are reported as median (interquartile range). A brain connectivity analysis was conducted for all EEG frequencies. P-values < 0.05 were considered statistically significant. All statistical calculations were done using GraphPad Prism version 9.5.1. This study was hypothesis-generating and ancillary to a pilot study investigating the effects of phrenic nerve stimulation on lung mechanics. Therefore, no a priori sample size estimation was made, and all patients with EEG and fNIRS data were eligible.

### Reporting summary

Further information on research design is available in the Nature Portfolio Reporting Summary linked to this article.

## Results

### Study patients

A total of 6 patients were included between 01 January 2022 and 31 December 2022, and completed the study as planned. Patient characteristics are presented in Supplementary Table 1. The median (inter-quartile range [IQR]) $PaO_2/FiO_2$ ratio was 161 (147–185) mmHg. Of note, one patient was admitted for an out-of-hospital cardiac arrest (Patient #1) and was diagnosed with acute hypoxic-ischemic encephalopathy 3 days after being enrolled in the study. In this patient, with known cerebral damage, brain responses to phrenic stimulation were expected to be absent or blunted compared to the responses observed in patients with an intact brain, and therefore constituted a possible "negative control", or "partially negative control". For this reason, the data from Patient #1 are presented separately from group data in Fig. 1, Fig. 2, and Fig. 3 (red dots). Due to a technical issue, EEG data from Patient #4 were not available. Therefore, we report EEG data from five patients and fNIRS data from six patients. All phrenic stimulation sessions were successfully conducted, and there were no adverse events.

### Medication, gas exchange and hemodynamics during the protocol

No changes in medication (sedative, norepinephrine) and $FiO_2$ were needed or recorded during the four sessions of the protocol. At enrollment, median (IQR) $FiO_2$ was 50 (40–60)%, pH 7.47 (7.44–7.47), $PaO_2$ 84 (69–100) mmHg, and $PaCO_2$ 41 (41–43) mmHg. Median (IQR) of the mean arterial pressure was 84 (74–95) mmHg for unpaced sessions and 84 (77–95) mmHg for paced sessions, $p = 0.88$ (Supplementary Table 2). Median (IQR) HR was 75 bpm (61–91) for unpaced sessions and 78 bpm (63–88) for paced sessions, $p = 0.42$ (Supplementary Table 2). Paced sessions showed a slight increase in cardiac index compared to unpaced sessions: median (IQR) 2.66 (2.36–3.13) $L/min/m^2$ vs. 2.56 (2.17–3.09) $L/min/m^2$, $p = 0.03$.

### Power spectral density EEG analyses

Gamma frequency (30–100 Hz). A significantly greater gamma power spectrum density (PSD) was observed bilaterally in the projections of the medial prefrontal cortex, insula, angular gyrus, and midline of the posterior cingulate gyrus during paced sessions compared to unpaced sessions (Fig. 1 and Supplementary Data File). This effect was observed in 10 out of 28 electrodes (seven brain regions) located in the frontal-temporal-parietal cortical regions (Supplementary Data File). In the subgroup

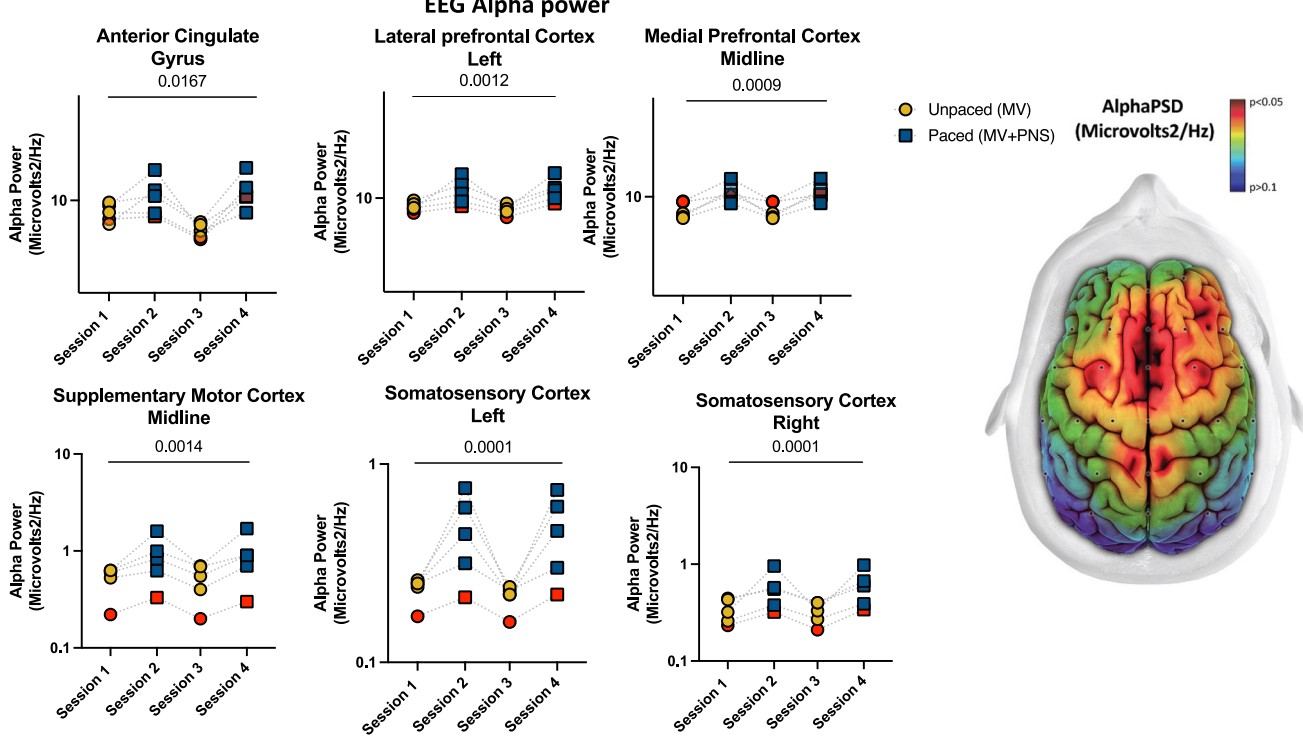

**Fig. 2 | Phrenic nerve increases EEG alpha frequency in deeply sedated mechanically ventilated patients.** EEG alpha power differences between paced and unpaced sessions: Dot plots showing the results of the linear analysis where statistically significant differences between the sessions for alpha frequency (8–13 Hz) were observed. The yellow and blue colors represent unpaced (MV, mechanical ventilation) and paced (MV + PNS, mechanical ventilation plus phrenic nerve stimulation) sessions respectively and the red colour represents data from Patient #1 who had an out-of-the-hospital cardiac arrest and was diagnosed with acute hypoxic-ischemic encephalopathy (a possible negative control due to an impaired brain-lung crosstalk). On the right, a brain topoplot displays the anatomical brain regions where a statistically significant change was observed between paced vs unpaced sessions. The red color in the topoplot indicates statistical significance defined as $p < 0.05$. The gradient of colors from yellow to dark blue indicates $p$-values greater than 0.05. (Source data file in Supplementary Data 2).

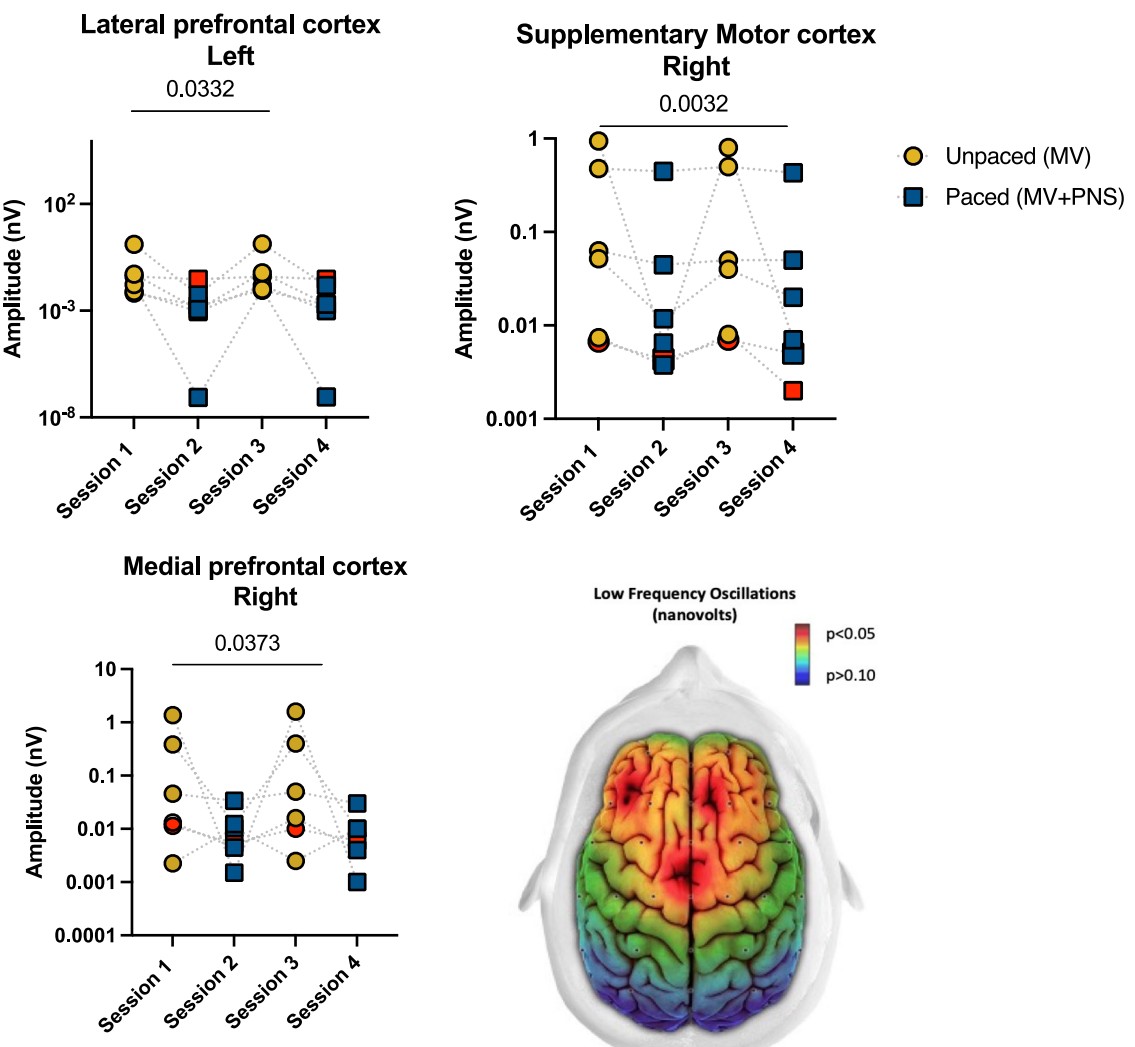

**Fig. 3 | Phrenic nerve changes the cerebral vessel tonus in deeply sedated mechanically ventilated patients.** fNIRS differences between pace and unpaced sessions: Dot plots show the results of the linear analysis where statistically significant differences between the sessions for functional near-infrared system data were observed. The yellow and blue colors represent unpaced (MV, mechanical ventilation) and paced (MV + PNS, mechanical ventilation plus phrenic nerve stimulation) sessions respectively and the red colour represents data from Patient #1 who had an out-of-the-hospital cardiac arrest and was diagnosed with acute hypoxic-ischemic encephalopathy (a possible negative control). On the bottom right, a brain topoplot displays the anatomical brain regions where a statistically significant change was observed between paced vs. unpaced sessions. The red color in the topoplot indicates statistical significance defined as $p < 0.05$. The gradient of colors from yellow to dark blue indicates p-values greater than 0.05. (Source data file in Supplementary Data 3).

analysis, the drug-addicted group had a greater proportional increase in brain activity in the left angular gyrus compared to the other subgroups, $p = 0.04$. Also, a tendency to significance for a greater proportional increase in the magnitude of the effect of phrenic nerve stimulation on the right angular gyrus was observed in the drug-addicted group compared to the other groups, $p = 0.09$.

**Beta (13–30 Hz).** No statistically significant change was observed between the sessions (Supplementary Data File).

**Alpha (8–12 Hz).** A significantly greater alpha PSD was observed in the medial prefrontal cortex, left lateral prefrontal cortex, supplementary motor cortex, diaphragm somatosensory cortex bilaterally, and anterior cingulate gyrus during paced sessions compared to unpaced sessions (Fig. 2 and Supplementary Data File). This effect was observed in 7 out of

28 electrodes (six brain regions) located in the frontal-parietal cortical regions (Supplementary Data File). In the subgroup analysis, the drug-addicted group had a greater proportional increase in brain activity in the supplementary motor cortex at midline and in the left diaphragm somatomotor cortex compared to the other subgroups, $p = 0.02$ and $p = 0.04$, respectively.

**Theta (4–8 Hz).** No statistically significant change was observed between the sessions (Supplementary Data File).

**Delta (0.5–4 Hz).** Delta PSD was significantly greater only in the medial prefrontal cortex (midline) in paced sessions compared to unpaced sessions (Supplementary Fig. 1). No statistically significant difference in PSD was observed between sessions in either the temporal or the parietal lobes.

# Brain Connectivity

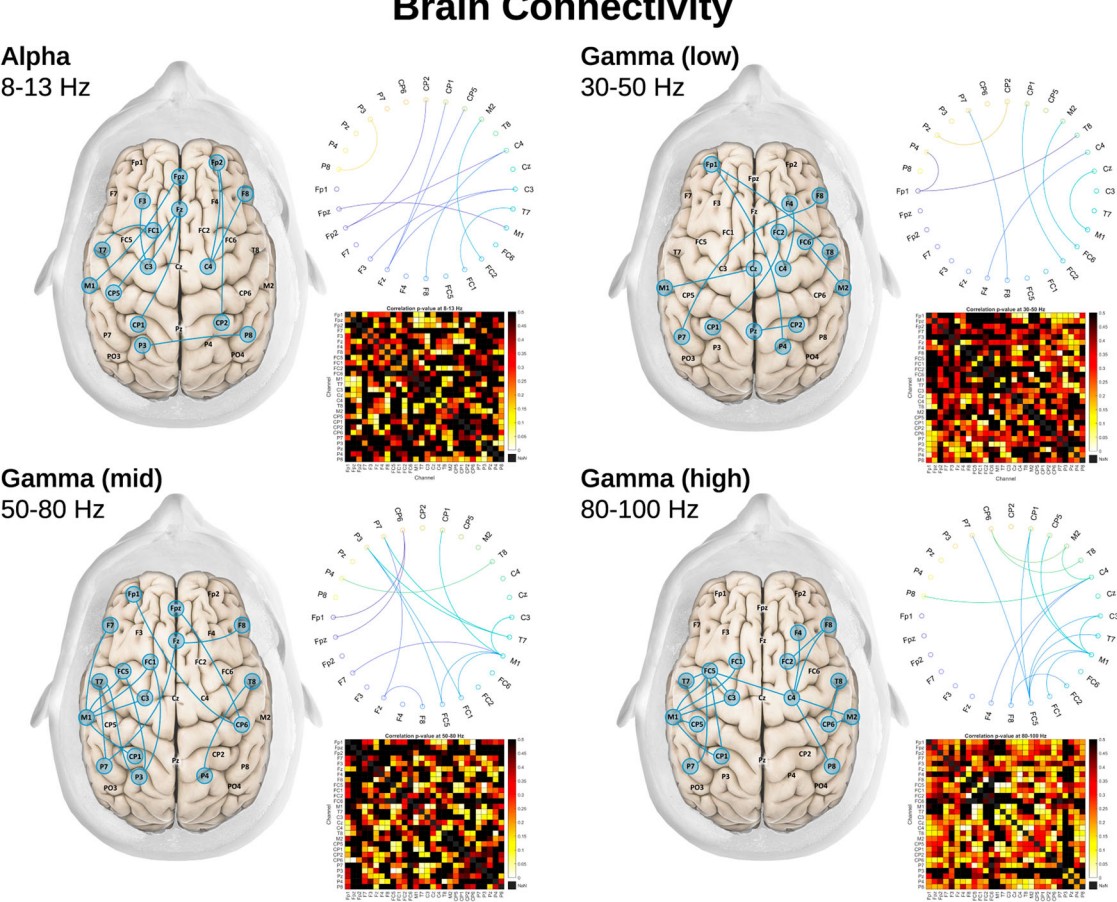

**Fig. 4 | Phrenic nerve increases brain connectivity in deeply sedated mechanically ventilated patients.** Brain connectivity analysis. Heatmaps displaying all the p-values for the correlation analyses between the paced (MV + PNS, mechanical ventilation plus phrenic nerve stimulation) and unpaced (MV, mechanical ventilation) sessions for the alpha frequency (8–13 Hz), low gamma frequency (30–50 Hz), mid-gamma frequency (50–80 Hz), and high gamma frequency (80–100 Hz). Brain connectograms on the left-hand side and circular graphs above the heatmaps show the brain areas where a statistically considerable difference in connection was observed in paced sessions compared to unpaced sessions. The dark red color in the heatmap indicates statistical significance defined as $p < 0.05$. The gradient of colors from light red to yellow indicates p-values greater than 0.05. (Source data file in Supplementary Data 4).

## Brain connectivity analysis from EEG signals

Greater frontal-temporal-parietal connectivity was observed during paced sessions compared to unpaced sessions (Fig. 4). This pertained to: i) the 30–50 Hz gamma frequency band, with increased connectivity between the right prefrontal cortex and right insula, left insula and anterior cingulate gyrus, left prefrontal cortex and right angular gyrus, and right supplementary motor cortex and right prefrontal cortex; ii) the 50–80 Hz gamma frequency band, with increased connectivity between right prefrontal cortex and anterior cingulate, left prefrontal cortex and left insula, left insula and left angular gyrus, left prefrontal and right angular gyrus, left insula and anterior cingulate gyrus, and left supplementary motor cortex and left insula; iii) the 80–100 Hz gamma frequency band, with increased connectivity between the right prefrontal cortex and right insula, left prefrontal cortex and left insula, left insula and left angular gyrus, right supplementary motor cortex and right insula, left supplementary motor cortex and left insula; iv) the alpha frequency, with increased connectivity between the right prefrontal cortex and right angular gyrus, left prefrontal cortex and left insula, right supplementary motor cortex and right prefrontal cortex, left supplementary motor cortex and left prefrontal cortex; v) the delta frequency, with increased connectivity between the left prefrontal cortex and left insula, and the right cortex and posterior cingulate gyrus (Supplementary Fig. 1).

## Analysis of fNIRS signals

We observed a significant reduction in the amplitude of LFOs in the sympathetic or myogenic range (0.08–0.13 Hz) during paced compared to the unpaced sessions. In this frequency range, median (IQR) oscillation amplitudes were lower during the paced sessions in the left lateral prefrontal cortex (0.0017 nV [0.0006–0.0058] vs. 0.0330 nV [0.0636–0.3612], $p = 0.03$), the midline supplementary motor cortex (0.0091 nV [0.0041–0.1449] vs. 0.0574 nV [0.0072–0.5923], $p = 0.01$), the right medial prefrontal cortex (0.0078 nV [0.0037–0.01765] vs. 0.0294 nV [0.0090–0.6311], $p = 0.04$) (Fig. 3). No statistically significant differences between sessions were observed in any brain regions in Patient #1. The global tissue saturation index was not significantly different between the unpaced (median [IQR] 57 [53–60]%) and paced (59 [55–60]%) sessions ($p = 0.31$). No statistical significance was found for LFO and VLFO amplitudes in the cardiac frequency range (1–1.25 Hz), the respiratory frequency range (0.25–0.18 Hz), the neurogenic or large vessels frequency range (0.05–0.07 Hz), and the metabolic or capillary frequency range (0.01–0.03 Hz). No statistically significant was observed for the proportional increase in brain activity between the subgroups during the phrenic nerve sessions.

## Discussion

This exploratory study showed significant changes in brain activity during phrenic nerve stimulation sessions in critically ill and deeply sedated, moderate ARDS patients undergoing MV, supporting the idea that it is possible to "wake the brain" of these patients at least to some extent, despite deep sedation. Among the significant changes observed were greater gamma and alpha power (in seven and six brain regions, respectively) and increased brain connectivity involving limbic areas. In addition to these EEG changes,

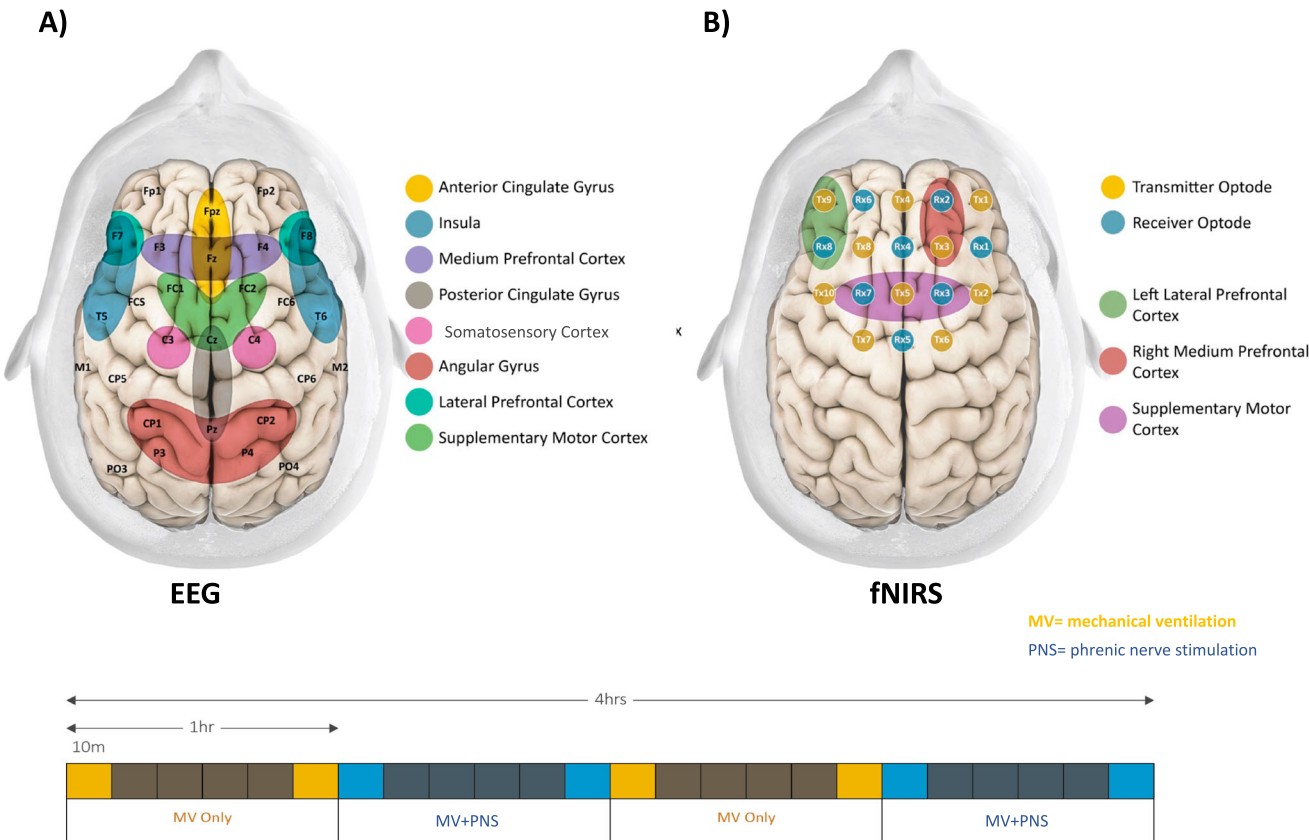

**Fig. 5 | EEG electrodes and fNIRS placement maps. A** Electrode placement for the electroencephalogram (EEG). **B** Optode placement for functional near-infrared spectroscopy (fNIRS). The average of the first 10 min and last 10 min in each session was pooled and used to compare EEG and fNIRS signals between the unpaced (MV, mechanical ventilation) and paced (MV + PNS, mechanical ventilation plus phrenic nerve stimulation) sessions.

our study also showed a reduction in the low-frequency oscillations of the fNIRS signal within the sympathetic range in three brain regions during the paced sessions. These observations align with the literature describing the effects of diaphragmatic breathing on brain function in awake humans, which shows increased activity, increased synchronization, and increased brain connectivity[18,19,37].

EEG power spectrum analysis provides a useful, well-known metric of the number of cortical neurons firing in synchrony within a determined brain area[38]. Every EEG frequency band is associated with a particular type of brain state or function[38]. For instance, greater gamma power has been linked to improved cognition and working memory than lower gamma power[38]. Recent studies have shown that breathing techniques, including diaphragmatic breathing, can deeply modulate neuronal activity and synaptic synchronization in awake humans[16–18,39]. Similar observations have also been made in neurosurgical patients during awake craniotomy[37,40]. Here, we show that the same type of effect can be obtained in response to electrical phrenic stimulation to artificially provoke diaphragm contractions. In our critically ill and deeply sedated ARDS patients undergoing invasive MV, we found greater gamma power and greater frontal-temporal-parietal connectivity during the paced sessions compared to the unpaced sessions. These changes involved the medial prefrontal cortexes bilaterally, the posterior cingulate cortex, the insula bilaterally, and the angular gyri bilaterally. All of these areas have been associated with diaphragmatic breathing–or phrenic stimulation–in prior studies[14,15,41]. Of note, gamma activity is normally absent or profoundly attenuated during deep sedation[42]. The fact that stimulating the phrenic nerve resulted in the presence of gamma activity in the frontal-temporal-parietal cortical regions of the limbic system in our patients therefore appears to be of particular importance[12,41,43]. The above observations suggest that phrenic stimulation

could be of interest as an approach to counteract the cognitive impairment and mood disorders frequently observed after sedation and MV in the ICU.

Experimental and clinical studies have shown that cortical inputs, including inputs from the prefrontal cortex and supplementary motor cortex, are essential in modulating the inspiratory effort adaptation during a mechanical inspiratory loading test[14,41,44–47]. Our study suggests that the phrenic nerve input to cortical areas might be a pivotal part of a common pathway for the activation of respiratory-related neural networks involving the prefrontal and supplementary cortical areas along with other brain areas. We observed greater alpha frequency power in the medial prefrontal cortex at midline, the left lateral prefrontal cortex, the supplementary motor cortex at midline, and the region of the somatosensory cortexes where diaphragm projections have been described[15]. These data align with the current literature describing brain responses to phrenic stimulation in awake healthy subjects[14,16,41]. More generally, we note that all the brain areas where we observed an increase in power spectral density have been associated directly or indirectly with the control and modulation of breathing[19].

In preclinical models, phrenic nerve stimulation has also been shown to increase heart rate variability, a surrogate of autonomic system activity[6,8]. This has been shown in pigs with healthy lungs and in pigs with moderate ARDS ventilated for 50 and 12 h, respectively, compared to pigs that received no phrenic nerve stimulation[6,8]. The fNIRS data collected in our patients, showing a phrenic stimulation-associated decrease in the low-frequency oscillations of the signal in the sympathetic or myogenic range within the right medial prefrontal cortex, left lateral prefrontal cortex and supplementary motor cortex at midline, suggest that phrenic nerve stimulation may modulate the autonomic system with an impact on brain vessel tone. This hypothesis derives from fNIRS studies that have shown changes in low-frequency amplitude oscillations were related to the

sympathetic tonus of cortical brain vessels[34,35], noting that brain vessels do not have parasympathetic innervation and cerebral autoregulation is exclusively regulated by the sympathetic system[34]. In this regard, there are anatomical communications between branches of the sympathetic system and the human phrenic nerve, which contains catecholaminergic fibres[48,49]. This makes phrenic stimulation-associated autonomic modulation of brain vessels tone plausible. We acknowledge that no mechanistic conclusions can be derived from our study, which was not designed for this purpose, however, we submit that our observations justify exploring this avenue.

The limitations of our study include its small sample size, and our findings will require corroboration on a larger scale. Also, the study is exploratory and descriptive. It does not provide mechanistic information and does not address the clinical implications of our findings. Future studies should be designed to investigate whether the various phrenic stimulation-associated brain changes that we describe could translate into improved cognitive performance, reduced incidence of delirium, or mitigation of mood disorders after MV and critical illness[19,38,43,50,51]. The impact of phrenic stimulation on dyspnea under MV should also be studied[46,47]. From a methodological point of view, it would be important to design control approaches, including the comparison of actual phrenic stimulation with sham stimulation, or the study of the brain effects of the stimulation of other nerve(s) (e.g., the vagus nerve). It would also be important to study control populations, such as non-sedated patients or patients with encephalopathies severe enough to alter or abolish somatosensory evoked potentials[38,52,53]. In this view, Patient #1 in our study is of interest, who, despite having suffered a hypoxic-ischemic brain insult, showed attenuated but not abolished brain responses to phrenic stimulation. Although the subgroup analysis showed that the phrenic nerve stimulation may have had an increased magnitude effect on the drug-addicted group, the sample size was too small sample to lead to definitive conclusions. Future studies should explore whether patient characteristics could modulate a greater magnitude effect of phrenic nerve stimulation.

Regardless of the limitations listed above, we believe that our study opens the possibility of a breakthrough in the field of VABI in the near future. Understanding the pathways involved in the brain-diaphragm-lung axis crosstalk and the alterations brought about by MV and sedation is essential, and the restoration of this crosstalk might prove highly beneficial. Such benefits have been suggested by preclinical studies, where hippocampus and brainstem inflammation were mitigated by phrenic nerve stimulation delivered by the same stimulator and therapy as used in our study[6,8,54].

In conclusion, adding phrenic stimulation on top of invasive mechanical ventilation in deeply sedated, critically ill, moderate ARDS patients increased cortical activity, connectivity, and synchronization. The observed changes resembled those during diaphragmatic breathing in awake humans. These results suggest that phrenic nerve stimulation has the potential to restore the brain-diaphragm-lung crosstalk when it has been shut down or impaired by MV and sedation. Future studies will be necessary to investigate whether our observed effects on brain excitability can indeed improve clinical outcomes and, for example, mitigate cognitive impairment and delirium in mechanically ventilated ARDS patients.

## Data availability
The datasets generated during and/or analysed during the current study are available from the corresponding author on reasonable request and cannot be freely available due to patient confidentiality. The source data for Fig. 1 is in Supplementary Data 1, Fig. 2 is in Supplementary Data 2, Fig. 3 is in Supplementary Data 3 and Fig. 4 is in Supplementary Data 4.

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

## Acknowledgements
We would like to thank Caro Minnick for helping with improving the quality of Figs. 1–5 and Andrew Lane (Lane Medical Writing), funded by Lungpacer Medical for English language and grammar editing. This work was funded by Lungpacer Medical Inc., Vancouver, Canada.

## Author contributions
T.B., E.R., S.R., A.D., T.S., and M.Dr. were responsible for hypothesis generation. T.B., E.R., S.R., A.D., T.S., and M.Dr. were responsible for the conception of this study. T.B., E.R., M.P., B.H., S.R., A.D., T.S., and M.Dr. contributed to the study design and data interpretation. T.B., E.R., S.R., A.D., T.S., B.H., M.P., J.M., M.De., J.L.H. and M.Dr. were responsible for writing the article. T.B., E.R., B.H., M.P., M.De., and M.Dr. performed data acquisition. T.B., M.P., E.R., B.H., T.S., and M.Dr. conducted data analysis. All authors approved the final version of the manuscript before submission.

## Competing Interests
The authors declare the following competing interests: T.B.: received a salary from Lungpacer Medical Inc. E.R.: consultant for Lungpacer Medical Inc. B.H.: consultant for Lungpacer Medical Inc. S.R.: co-inventor and received personal fees from Lungpacer Medical Inc., Vancouver, Canada A.D.: Medtronic, grants, personal fees and nonfinancial support from Philips, personal fees from Baxter, personal fees from Hamilton, personal fees and non-financial support from Fisher & Paykel, grants from French Ministry of Health, personal fees from Getinge, grants and personal fees from Respinor, grants and nonfinancial support from Lungpacer, outside the submitted work T.S.: reports personal fees for consulting and teaching activities from AstraZeneca France, Chiesi France, KPL consulting, Lungpacer Inc., OSO-AI, TEVA France, Vitalaire. He is a stock shareholder of startups Hephaï and Austral Dx. He is listed as inventor on issued patents (WO2008006963A3, WO2012004534A1, WO2013164462A1) describing EEG responses to experimental and clinical dyspnea. MDr: received personal fees from

 9

Lungpacer Medical Inc., Vancouver, Canada and was a member of the Clinical Advisory Board of Lungpacer Medical Inc., Vancouver, Canada. M.P., J.M., M.D., and J.H. declare no competing interests.
