## [Peer Review File · Communications Medicine]

Reviewers' comments:

Reviewer #1 (Remarks to the Author):

This work examines whether phrenic nerve stimulation can enhance brain activity and connectivity, in patients with acute respiratory distress syndrome who are deeply sedated and mechanically ventilated. This could reduce the side effects which can occur from mechanical ventilation.

This is a well written manuscript and provides all the required information for the study. The sample size is small, and the authors acknowledge this and are reasonable in the discussion in the results and the conclusions. The methodology for the work is all appropriate. Further work will be required to made a definitive assessment of the effectiveness of this treatment.

Reviewer #2 (Remarks to the Author):

Department of Anaesthesia and Intensive Care Medicine,
Beaumont Hospital,
Dublin 9.

Re: Waking the Brain in Sedated Mechanically Ventilated Acute Respiratory Distress Syndrome Patients with Phrenic Nerve stimulation.

Dear Editor,

Thank you for asking me to review this interesting paper. I am familiar with Professor Martin Dres's work with the lungpacer system.

This is an interesting paper, it is well written and it builds on works previously published by the same authors. Also of note is Professor Dres's insight piece from Intensive Care Medicine (Intensive Care Medicine (2022) 48; 1299-1301) and equally the authors have referenced Albccieta GM's paper from the BJA.

Professor Dres is a field leader in the areas of ventilator induced diaphragmatic dysfunction, phrenic ventilation and its potential applications. His work presents ground-breaking approaches for the future. I strongly believe the hypothesis that phrenic ventilation can ameliorate the damaging effects of positive pressure ventilation.

Equally we know from sedation studies, "SPICE" and others that excessive sedation can be extremely damaging with prolonged ventilation, delirium and cognitive deficits for some time post ICU discharge. We know from Margaret Herridge's work how outcomes in critical illness may be influenced by what we do as clinicians, her recommendation: adherence to ABCDEF bundles and mitigation of iatrogenesis are key messages. (N Engl J Med 2023;388:913-24.)

Typical of Industry sponsored studies I note most authors are employed or subsidised in some way by Lungpacer limited. Regardless of sponsorship, the manuscript is excellent.

To the current study, very well written and presented and I believe with minor amendments only it fully deserves publication. I think the results are quite impressive however I am just not sure how these specific endpoints will translate to real world endpoints. The authors do acknowledge their limited sample size and equally limited periods of mechanical ventilation augmented with phrenic stimulation.

Each of the 6 patients received 2 periods of mechanical ventilation plus phrenic stimulation interspersed with 2 periods standard which led to increased cross-talk within the brain. Given the degree and complexity of the neuromonitoring involved the limited sample size is completely understandable.

Apart from sample size and limited periods of phrenic stimulation both of which are completely understandable given the complexity of neuromonitoring employed, I do have some minor criticisms, each which may be addressed very easily.

First I note that the method's section extends from Lines 237 – 363, it seems to be out of position within the manuscript and does not follow the introduction. This will likely be addressed pre-publication.

The supplementary file clinical investigation plan is very comprehensive however but it related to the paper from Anesthesiology than the current study (PIRAT study).

Spelling check line 246. extra "s"

Line 368 incomplete figures xxx

P values within the manuscript are reported to four decimal points, is this necessary, I would favour 2.

To conclude, well written and clear manuscript, fully deserves publication.

Many thanks

James O'Rourke

Reviewer #3 (Remarks to the Author):

In this study by Bassi et al., the authors undertook an exploratory study based on available literature to investigate whether the phrenic nerve stimulation could improve the brain connectivity and respiratory outcomes in waking the brain from deep sedation and mechanical ventilation stage in patients with severe respiratory distresses. While the study revealed important clinical findings in the context of the phrenic nerve stimulation therapy in respiratory distressed patients, the study suffers from certain drawbacks that make it unacceptable in its present form.

The authors may consider the following comments to further the scientific soundness of this study:

1) In patient characteristics table S1, please include the following points:

a) Ethnic background of these patients.

b) Treatment duration and any follow-up observations, if conducted.

c) Reference ranges for parameters - weight, Pa2/FiO2, FiO2, Tidal vol based on the age, sex, and ethnicity.

2) Include the patient selection and exclusion criteria.

3) Compare changes in examined parameters in 6 patients based on the reference values of healthy controls before and after administering pacing therapy to reveal the fold-change of restoration of relevant parameters, if there are any.

4) A subgroup analysis of these parameters may be helpful in interpreting the study results in a better way. The patient #3 and #5 can be combined in the drug addiction subgroup; patient #2 and #6 in the non-addicted, and patient #1 and #4 respectively in the HIE and epileptic subgroups.

5) Did the authors performed any cognitive battery tests after stabilizing the patient following pacing treatment and before discharge?

6) Was there any self or family reported memory and/or cognitive problems reported on admission?

7) Please include the statement whether this study was approved by the IRB, if yes, please include the study reference number.

Please mention if these patients proved written informed consent for this publication.

8) Was this a single or multi-center study? What was the study location?

Point-by-point

REVIEWER #1

REVIEWER COMMENT #1: Further work will be required to make a definitive assessment of the effectiveness of this treatment.

RESPONSE: Thank you very much for your comments. We agree that further work will be required to make a definitive assessment of the effectiveness of this treatment.

REVIEWER #2

REVIEWER COMMENT #1: First I note that the method's section extends from Lines 237 – 363, it seems to be out of position within the manuscript and does not follow the introduction.

RESPONSE: Thank you very much for pointing this out. We have moved the methods section between the Introduction and Results sections.

REVIEWER COMMENT #2: The supplementary file clinical investigation plan is very comprehensive however but it related to the paper from Anesthesiology than the current study (PIRAT study).

RESPONSE: Thank you very much for highlighting this issue. The protocol from the PIRAT study was uploaded upon the editor's request to show that Revision C of the study protocol was amended and approved in February 2022.

REVIEWER COMMENT #3: Spelling check line 246. extra "s"

RESPONSE: Thank you very much for catching an extra "s" in line 246. We deleted it.

REVIEWER COMMENT #4: Line 368 incomplete figures xxx

RESPONSE: Thank you for highlighting our oversight. We add the figure's numbers.

REVIEWER COMMENT #5: P values within the manuscript are reported to four decimal points, is this necessary, I would favor 2.

RESPONSE: Thank you for the suggestion. Now we are reporting p-values with two decimals, however in the figures and Table in the supplement we are reporting in four decimals due to small p-values observed in some interactions.

REVIEWER #3

REVIEWER COMMENT #1: In patient characteristics table S1, please include the following points:

- a) Ethnic background of these patients.
- b) Treatment duration and any follow-up observations, if conducted.
- c) Reference ranges for parameters - weight, Pa2/FiO2, FiO2, Tidal vol based on age, sex, and ethnicity.

RESPONSE: Thank you very much for suggesting adding these variables. We agree with adding them in Table S1 except ethnic background which is a variable that cannot be legally collected in France. Please see Table S1 in the Supplement.

REVIEWER COMMENT #2: Include the patient selection and exclusion criteria.

RESPONSE: Thank you for highlighting this issue. In the original submission we wrote the inclusion and exclusion criteria in the methods section. We highlight in yellow the criteria in lines 81-94. Please see below.

“Inclusion criteria were patients with moderate ARDS according to the Berlin definition⁴⁶, receiving invasive MV for at least 48 hours but no longer than 5 days, and receiving continuous sedation, with Richmond Agitation Sedation Scale (RASS) \leq -3. Non-inclusion criteria were: septic shock with hemodynamic instability (requiring norepinephrine or epinephrine >0.5 $\mu\text{g}/\text{kg}/\text{min}$), impossible access to the left subclavian vein, use of

neuromuscular blocking agents within the last 12 hours, bacteremia documented within the last 48 hours or uncontrolled source of infection, ongoing extracorporeal lung circulation, enrolled in other studies of an investigational drug or device that could have affected the outcomes of the current study, pre-existing neurological, neuromuscular or muscular disorder that could have affected the respiratory muscles, body mass index >45 kg/m², known or suspected phrenic nerve paralysis, any electrical device (implanted or external) that could have been prone to interaction with or interference from the phrenic stimulation catheter, including neurological pacing/stimulator devices and cardiac pacemakers and defibrillators, no affiliation to the French health insurance system, curatorship, imprisoned, known or suspected to be pregnant, or lactating.”

REVIEWER COMMENT #3: Compare changes in examined parameters in 6 patients based on the reference values of healthy controls before and after administering pacing therapy to reveal the fold-change of restoration of relevant parameters, if there are any.

RESPONSE: This is a great suggestion. Deep sedation and mechanical ventilation may play an important role in modulating cortical activity which could be a major confounder when we compare healthy subjects with deeply sedated mechanically ventilated patients. Also, because of the potential high interindividual variability, it is not advisable to compare the values between subjects. The cross-over design assisted us in pointing out the cortical changes secondary to phrenic nerve stimulation (the only variable that changed during the sessions) adding a robust argument for the effect of the afferent phrenic nerve signal in modulating cortical activity. We will consider comparing healthy subjects with deeply sedated mechanically ventilated patients in the design of future studies.

REVIEWER COMMENT #4: A subgroup analysis of these parameters may be helpful in interpreting the study results in a better way. The patient #3 and #5 can be combined in the drug addiction subgroup; patient #2 and #6 in the non-addicted, and patient #1 and #4 respectively in the HIE and epileptic subgroups.

RESPONSE: Thank you for the comment! We agree on adding a subgroup analysis to explore whether similar patient characteristics could modulate the magnitude of the effect observed during the phrenic nerve stimulation sessions. However, we divided into three subgroups (HIE, non-drug addicted, and drug addicted), instead of four subgroups. This is because the epileptic patient enrolled in our study was in a no-status epilepticus and was admitted due to broncho-aspiration. Thus, we decided to consider this patient in the non-drug addicted group. Please see lines 191-195, 231-235, 243-245, 282-283, and 357-360. Please see below.

Lines 191-195:

“A subgroup analysis using a mixed-effect analysis of the variance of the magnitude effect of the phrenic nerve stimulation on the brain was conducted to investigate whether patients’ characteristics could modulate the magnitude of the observed effect during the phrenic nerve stimulation sessions. Patients were divided into three subgroups, hypoxic-ischemic (#1) encephalopathy, non-drug addicted (#2, #4, #6), and drug addicted (#3, #5).”

Lines 231-235:

“In the subgroup analysis, the drug-addicted group had a greater proportional increase in brain activity in the left angular gyrus compared to the other subgroups, $p=0.04$. Also, a tendency to significance for a greater proportional increase in the magnitude of the effect of phrenic nerve stimulation on the right angular gyrus was observed in the drug-addicted group compared to the other groups, $p=0.09$.”

Lines 243-245:

“In the subgroup analysis, the drug-addicted group had a greater proportional increase in brain activity in the supplementary motor cortex at midline and in the left diaphragm somatomotor cortex compared to the other subgroups, $p=0.02$ and $p=0.04$, respectively.”

Lines 282-283:

“No statistically significant was observed for the proportional increase in brain activity between the subgroups during the phrenic nerve sessions.”

Lines 357-360:

“Although the subgroup analysis showed that the phrenic nerve stimulation may have had an increased magnitude effect on the drug-addicted group, the sample size was a too small sample to lead to definitive conclusions. Future studies should explore whether patient characteristics could modulate a greater magnitude effect of phrenic nerve stimulation.”

REVIEWER COMMENT #5: Did the authors performed any cognitive battery tests after stabilizing the patient following pacing treatment and before discharge?

RESPONSE: This is an outstanding idea. However, we did not perform any cognitive battery tests after stabilizing the patient following phrenic nerve stimulation because it was not part of the study protocol.

REVIEWER COMMENT #6: Was there any self or family reported memory and/or cognitive problems reported on admission?

RESPONSE: Again, great idea, but unfortunately, we have not collected any self or family-reported memory and/or cognitive problems reported on admission.

REVIEWER COMMENT #7: Please include the statement whether this study was approved by the IRB, if yes, please include the study reference number. Please mention if these patients proved written informed consent for this publication.

RESPONSE: Thank you very much for pointing out this issue. We have added the IRB number (lines 69-75) and a statement about informed consent on lines 78-79 that reads now as follows:

Lines 69-75:

“This report is an ancillary investigation from a cross-sectional single-center study (ClinTrials.gov: NCT04844892) approved by the ethics committee (Comité de Protection de Personnes (CPP), Ile-De-France X, Paris, France) on June, 15th, 2021. The original study was designed to investigate the effects on lung mechanics of phrenic nerve stimulation in critically ill mechanically ventilated patients. The results (lung mechanics,

hemodynamics and gas exchange) have been published elsewhere²⁵ and the present study reports unpublished data of a subgroup of patients whose brain data were collected after an amendment (21.04819.492021-MS01) to the primary study was approved on February, 15th, 2022 by the same ethics committee.”

Lines 78-79:

“A signed informed consent was obtained from the substitute decision-makers for each patient prior to study inclusion.”

REVIEWER COMMENT #8: Was this a single or multi-center study? What was the study location?

RESPONSE: It was a single center and also, and we highlighted the location where the study was conducted. We added this information on lines 69-71 (see below).

Lines 69-71:

“This report is an ancillary investigation from a cross-sectional single-center study (ClinTrials.gov: NCT04844892) approved by the ethics committee (Comité de Protection de Personnes (CPP), Ile-De-France X, Paris, France) on June, 15th, 2021.”

REVIEWERS' COMMENTS:

Reviewer #2 (Remarks to the Author):

Dear Dr Barnes,

I have reviewed the authors rebuttal letter and read through their manuscript again, I am satisfied that each of my concerns have been addressed thoroughly. If I could make one small comment Line 83 - Non-Inclusion Criteria suggest change to Exclusion criteria.

This article is very well written and I would fully support its publication in its current (revised) format.

Many thanks for asking me to review

Yours Sincerely

Dr James O'Rourke

Reviewer #3 (Remarks to the Author):

The authors have satisfactorily answered to all comments and made the required modifications and corrections to the revised manuscript. I would recommend for the acceptance of this manuscript.

Point-by-point

REVIEWER #2

REVIEWER COMMENT #1: If I could make one small comment Line 83 - Non-Inclusion Criteria suggest change to Exclusion criteria.

RESPONSE: Thank you very much for your suggestion. We have changed the wording to exclusion criteria as suggested.

REVIEWER #3

REVIEWER COMMENT #1: The authors have satisfactorily answered to all comments and made the required modifications and corrections to the revised manuscript. I would recommend for the acceptance of this manuscript.

RESPONSE: Thank you very much for all your input and feedback.